# Pressure and bias dependence of the rate-limiting steps of the oxygen reduction reaction

Alex Ricardo Silva Olaya [1,2], Jody Druce[1,2], Jose M. Gisbert-Gonzalez[1], Eduardo Ortega [1], Beatriz Roldan Cuenya [1] & Sebastian Z. Oener [1] ✉

The oxygen reduction reaction limits the energy efficiency of $H_2$ fuel cells and Li-air batteries, yet, it remains poorly understood within popular kinetic frameworks. Here, we study the oxygen reduction reaction on Pt/C, Ir/C, Ru/C and Rh/C nanoparticles as a function of electrochemical bias, temperature and $O_2$ pressure at industrially-relevant conditions in membrane electrode assemblies. Bias-, and pressure-dependent Arrhenius analysis reveals distinct changes in the (apparent) activation energy and pre-exponential factor that we relate to kinetics that cascade through a series of rate-limiting steps and transition states. Further, while the kinetics are accelerated by the pressure and bias, they remain pinned to pseudo-capacitive reduction processes and structural changes at the water-solid interface. Collectively, our study informs on how the free energy driving force and pressure tune the degree of rate control of rate-limiting steps and transition states of (electro)catalytic multi-step reactions and how this is related to structural and chemical changes at the interface. This is at the very heart of catalysis.

The oxygen reduction reaction (ORR) is one of the most practically-relevant electrocatalytic reactions of the green economy, as it underpins both $H_2$ fuel cells and metal-air batteries. However, due to the complexity of the multi-step reaction, the ORR remains poorly understood, limiting knowledge-driven catalyst design for more energy-efficient systems.

The complete reduction of $O_2$ requires a multi-step sequence[1], with distinct intermediates forming during each step in acidic ($O_2 + 4H^+ + 4e^- \rightarrow 2H_2O$) or alkaline media ($O_2 + 2H_2O + 4e^- \rightarrow 4OH^-$)[2]. ORR kinetics are widely modeled with outer-sphere-type rate equations, that assume that the "intrinsic activity" is governed by a constant exchange current density, $j_0$, and that the applied free energy difference (overpotential) simply reduces the activation energy over the whole overpotential range. Related, the Bell-Evans-Polanyi (BEP) relationships[3–8] explain activity differences for a set of catalysts to activation energy differences, keeping the activation entropy constant. Further, it is still often assumed that catalyst activity for a multi-step

reaction can be reduced to one rate-determining intermediate. However, all of the above find their origin in bulk redox reactions[9]. For inner-sphere reactivity at heterogenous interfaces, the applied free energy can influence: (i) ion transport across the electric-field-dependent double layer, including the removal or acquisition of a solvation shell; (ii) the binding energy and surface coverage of reaction intermediates; and (iii) structural or chemical changes at the surface or within the solvent. Currently, there is no comprehensive view how these processes impact the (apparent) activation parameters and, thus, electrocatalyst kinetics.

Previous Arrhenius analysis for the ORR has highlighted that catalyst surfaces are not static. For example, a bias-dependent Arrhenius pre-exponential factor was observed for Pt single crystals in alkaline media and rationalized via a bias-dependent surface coverage of $O_2$ and its intermediates that compete with $OH_{(ads)}$ for active sites[10,11], and similar conclusion were drawn for supported Pt nanoparticles and Pt-Nafion dispersions in acid at low current densities[12,13]. Further, a

[1]Department of Interface Science, Fritz-Haber Institute of the Max Planck Society, Berlin, Germany. [2]These authors contributed equally: Alex Ricardo Silva Olaya, Jody Druce. ✉e-mail: oener@fhi-berlin.mpg.de

changing pre-exponential factor was linked to a bias-dependent activation entropy that could impact interfacial solvation in the double layer[14,15], analogous to the findings by Conway for the hydrogen evolution reaction (HER) on sp-metals[16–19]. More recently, we observed extended compensation between the pre-exponential factor and apparent activation energy for the ORR and other reactions at low current densities[20,21] and under high mass transport conditions for isolated solvation kinetics in bipolar membranes[22] and a large set of HER catalysts in hydrogen pump cells[23]. However, for the ORR, previous Arrhenius studies, including our own[12–15,20], were limited to low current densities, potentials and conducted almost exclusively at low partial $O_2$ pressures.

At the triple-phase boundary in electrocatalysis, the adsorption of gaseous reactants is frequently implicated in rate-limiting steps, such as during $CO_2$, $N_2$ and $O_2$ reduction. Thus, the pressure-dependent chemical potential changes in the transition state complex are of broad interest. Despite this, the kinetic effect of polarizing catalyst surfaces via the reactant pressure is poorly addressed in heterogenous catalysis. Often, the effect of the pressure is exclusively relegated to an increasing concentration term in the rate law, e.g. $\nu = \left[O_2\right]^n k$, depending on the reaction order (n), instead of fundamental pressure-dependent change in $k$.

For electrocatalysis, this short-coming is in part caused by the popularity of liquid electrolyte cells, which are quickly impacted by mass transport due to the low solubility of gas in liquids[24], including for high spin speeds in rotating disk electrodes (RDEs) with pressure control[25]. In summary, decades of fundamental catalyst research have led to a very partial picture of the ORR, limited to conditions close to electrochemical equilibrium and far removed from any industrially-relevant rates. In contrast, the triple-phase boundary in gas diffusion electrodes affords access to much higher current densities, potentials and pressures.

Here, we comprehensively study the temperature-, $O_2$-pressure- and bias-dependent ORR kinetics across nanoparticle catalysts in membrane-electrode assembly (MEA) fuel cells. We observe that the overpotential-dependent Arrhenius pre-exponential factor and apparent activation energy cascade through a series of rate-limiting steps and, potentially, transition states with varying degrees of rate control, which are closely linked to and limited by pseudo-capacitive processes at the solid-water interface.

## Results

We use an $H_2$ fuel cell that operates the ORR on a low-loading (~0.2 mg cm$^{-2}$) Pt/C, Ir/C, Ru/C or Rh/C gas diffusion electrode (GDE). The GDE is charge-balanced by the facile hydrogen oxidation reaction (HOR) on a higher-loaded (~2 mg cm$^{-2}$) Pt/C GDE, which also serves as an internal reference under well-controlled $H_2$ pressure. Conversely, the total cell overpotential informs directly on the overpotential of the opposing, low-loading ORR GDE when corrected for the small contribution of the HOR, especially, at higher current densities (Supplementary Fig. 1). Both electrodes are separated by proton conducting Nafion (50 μm) in a membrane electrode assembly (MEA). All potentials are corrected for the ohmic drop (IR) (Supplementary Fig. 2). Steady-state ORR rates are recorded by controlling the overpotential (analogously referred to as the bias here), gas flow and temperature (25 °C–45 °C in 5 °C increments). The $O_2$ pressure is changed (2–6 bar in 1 bar increments), while keeping the $H_2$ pressure in the compartment of the internal reference electrode constant. $H_2$ cross-over was evaluated before each measurement from the Faradaic currents in opposite polarity at low overpotentials, which are caused by the HOR instead of ORR. However, within the tested temperature and pressure ranges, the Nafion leads to minimal $H_2$ cross-over (Supplementary Fig. 3). To obtain stable ORR rates, the cells were first conditioned for 4 h followed by repeated cyclic voltammograms (Supplementary Fig. 4). Next, multi-step chronoamperometry was performed with

individual potential steps of 60 s, ensuring steady-state ORR currents with negligible pseudo-capacitive charging (Supplementary Fig. 5). Electron microscopy of Pt/C and Rh/C nanoparticles carried out before and after the reaction, including at elevated pressure up to 6 bar indicate that particle size, shape and dispersion do not drastically change during the reaction (Supplementary Figs. 6–11). These results are consistent with a total change of the nanoparticle surface area of 5–10%, as implied by hydrogen underpotential deposition and CO displacement before assembly, after conditioning and after 4 h operation (Supplementary Fig. 12). Finally, we tested for the impact of mass transport by operating the cells with reduced $O_2$ concentration in the gas feed and observe that the kinetics presented in the main are not dominated by mass transport, unless stated otherwise. More discussion later and in Supplementary Note 1 and Supplementary Figs. 13-14.

For the Arrhenius linear regression, the high mass transport and stability of the fuel cell provides very high $R^2$ coefficients (0.95-0.999) for the Pt/C, Rh/C, Ir/C and Ru/C (Supplementary Figs. 15–18). The lack of substantial non-linear Arrhenius behavior reduces the likelihood—within the studied temperature range—that significant temperature-dependent intermediate coverages, temperature-induced structural changes or a deformed transition state[26] lead to pronounced temperature-dependent activation energies. Note, this does not preclude the existence of temperature-dependent structural or chemical changes that are relevant on faster time scales at highly dynamic active sites or other effects that could become relevant in a larger temperature range. Here, we are focusing on the bias- and pressure-dependence of the apparent activation parameters that arise from ensemble properties. Further, while any temperature range in aqueous electrochemistry might appear small in the eyes of a thermo-catalysis audience, we note that an ORR study of Pt in concentrated phosphoric acid spanning a 200 °C also concluded that ORR kinetics are dominated by a bias-dependent entropy at low current densities[14].

High $R^2$ values and extended temperature ranges are generally important to exclude artefactual $\log A$-$E_A$ compensation along an isokinetic or average temperature that can arise from random[27] or systematic[28] errors for repeated measurements of individual $\log A$-$E_A$ pairs, as often observed in bio[29,30]- and thermal catalysis[31]. However, we observe incremental changes in the pre-factor and activation energy with incremental increase in electrochemical bias with high covariance, that we can directly link to precise changes in the current density. Therefore, especially the study of the systematic bias dependence of the apparent activation parameters is unique to electrochemistry and has been prominently pursued by Conway[16–19]. On the one hand, this enables us to study the susceptibility of the catalytic rate to a differential change in the free energy of one rate-limiting step or transition state (in a certain overpotential range) for relatively simple reactions such isolated ion solvation in bipolar membranes[22] or a dominant Volmer- or Tafel step in the HER[23]. Conversely, we observed almost "text-book" like Butler-Volmer HER kinetics for the fast Pt group metals in acid, especially when accounting for the reverse rate[23]. In contrast, for complex multi-step reactions, such as the OER[21] and ORR, the bias-dependent (apparent) activation parameters cannot be directly linked to the activation entropy and enthalpy of one rate-limiting transition state, unless one step is largely dominant in a certain overpotential range. In general, the bias-dependent changes in the apparent activation parameters allow us to study how individual steps change their degree of rate control[32,33] over a catalyst-solution interface that undergoes substantial changes with bias, including interfacial solvation, the formation of excess charge, intermediate coverages or free energy changes and catalyst restructuring. However, when combined with operando spectroscopy, a powerful tool emerges that can provide deeper insights into the kinetics, as we have recently started to explore for the OER[21].

Transition state theory originates from statistical distributions in energy and space, allowing the calculation and rationalization of

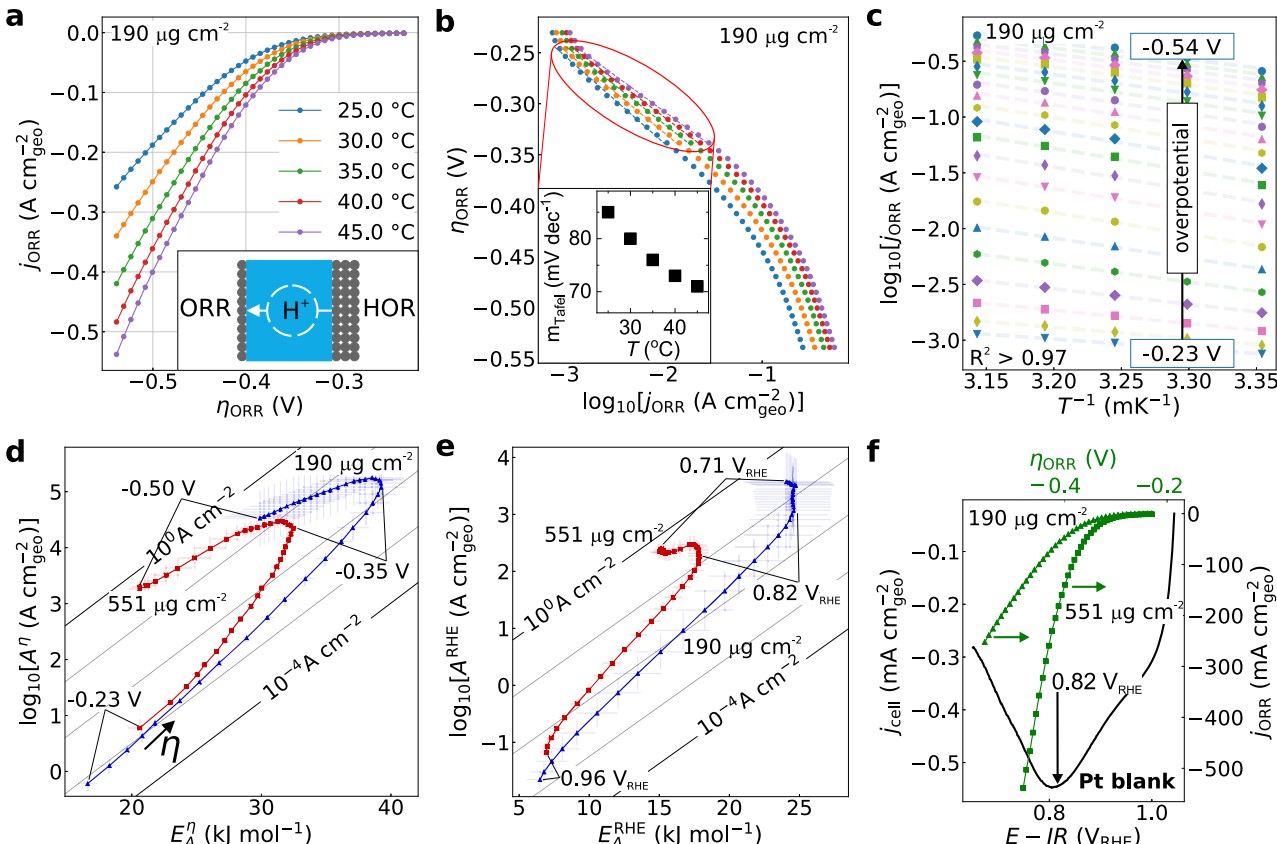

**Fig. 1 | Electrochemical Arrhenius analysis of oxygen reduction reaction (ORR) on Pt/C in Nafion-based membrane electrode assembly. a** ORR polarization curves between 25°C–45°C. The inset shows the membrane electrode assembly (MEA) with a low-loading ORR electrode and a high-loading counter and internal reference electrode that runs the facile hydrogen oxidation reaction (HOR) separated by an acidic proton ($H^+$) conducting membrane (blue). **b** Tafel plots with temperature-dependent Tafel slopes (inset). **c** Arrhenius analysis of the ORR for selected overpotentials, with an arrow indicating the direction of increasing overpotential. The loading for (**a**–**c**) is 190 µg cm$^{-2}$. **d** Kinetic map of the (real) activation energy, $E_A^\eta$, and the Arrhenius pre-exponential factor, $\log A^\eta$, as function of overpotential, $\eta$, for two loadings. Note, $E_A^\eta$ and $\log A^\eta$ increase with bias in the region of the linear Tafel slope in (**b**). **e** Kinetic map of the (formal) activation energy, $E_A^{RHE}$, and the Arrhenius pre-exponential factor, $\log A^{RHE}$, as function of absolute

potential, $E_{RHE}$, for two loadings. **f** Faradaic steady state ORR currents (green) from chronoamperometry at 45°C and cyclic voltammogram (black) of Pt/C in absence of $O_2$ with a scan rate of 10 mV s$^{-1}$ for two loadings. The reduction wave is maximized right at the potential where $E_A^{RHE}$ is maximized in Fig. 1e for both loadings, suggesting a critical influence of pseudo-capacitive processes at the water-solid interface in controlling the intrinsic catalyst activity. $\log A(\eta)$ and $E_A(\eta)$ were obtained from Arrhenius analysis, based on five temperatures. The error bars are based on linear regression standard error. The diagonal lines in the background are iso-current lines at 25 °C, i.e., pairs of $\log A$ and $E_A$ that result in the same current density. Pt mass-loadings of 190 µg cm$^{-2}$ and of 551 µg cm$^{-2}$ were used. All overpotentials are corrected for the temperature dependence of the ORR equilibrium potential (0.83 mV K$^{-1}$). All measurements were conducted in Nafion-based membrane electrode assemblies.

macroscopic observables from microscopic ensemble properties. Regardless of the fact that the parameters of transition state theory cannot be directly extracted from Arrhenius analysis for multi-step reactions[34], we can treat the ORR kinetics on the GDE as an idealized-triple-phase boundary arising from ensemble properties of highly-dynamic and structurally-complex active sites and their molecular environment. We make this assumption first on the basis of the very high rate with which our cells deliver $O_2$ gas, electrons and solvated protons, and remove $H_2O$ (see Supplementary Note 1-2), and second because we use only low-loading GDEs and obtain very high Arrhenius linear $R^2$ regression values (Supplementary Fig. 15-18).

All overpotentials are corrected for the temperature dependence of the ORR equilibrium potential (0.83 mV K$^{-1}$). This involves interpolation as outlined in Supplementary Fig. 19 and detailed in the Methods. Further, we distinguish[35] between the real activation parameters ($\log A^\eta$, $E_A^\eta$), obtained from Arrhenius analysis of currents at different overpotentials, $\eta$, and the formal activation parameters ($\log A^{RHE}$, $E_A^{RHE}$), obtained from Arrhenius analysis of currents at absolute potentials, $E_{RHE}$, referenced to the reversible hydrogen electrode (RHE) (Supplementary Note 3 and Supplementary Fig. 20).

Figure 1a shows the ORR polarization curves for the low-loading Pt/C GDE at five temperatures to overpotentials (current densities) up to -0.55 V ($\geq$ 500 mAcm$^{-2}$). In the following, we first discuss the results for the whole range, before limiting the overpotentials (current densities) to -0.45 V ($\leq$ 250 mAcm$^{-2}$) to exclude substantial impact of mass transport (Supplementary Figs. 13-14). Figure 1b shows a typical Tafel plot ($\log j$ vs.$\eta$) for each temperature, with slopes (inset) calculated at overpotentials between -0.25 V and -0.34 V, for comparison with literature RDE data in a similar range. For the impact of the overpotential correction on the Tafel slopes, see Supplementary Fig. 21. Figure 1c shows the Arrhenius linear regression for exemplary overpotentials and highlights that the lines do not converge to a common y-intercept. From the slope, $E_A^\eta$ can be determined, while extrapolation of $\log j$ to infinite temperatures provides $\log A^\eta$ at different overpotentials.

Figure 1d shows the kinetic map of $E_A^\eta$ and $\log A^\eta$ for two different loadings. In the following we focus on the main features of the kinetic maps before addressing the loading (in)dependencies separately. With negative overpotentials up to -0.35 V, $E_A^\eta$ and $\log A^\eta$ increase together and (partially) compensate. Notably, in this regime we observe linear Tafel slopes in Fig. 1b, yet, the rates are driven by the (over)

compensating $\log A^\eta$ and not a decreasing $E_A^\eta$. These results are a prime example of the importance of temperature-dependent electrochemistry. In particular, the temperature dependence of the Tafel slope ($b = RT\ln10/(F\alpha(T))$) has been interpreted by Conway via enthalpic ($\alpha_H$) and entropic ($\alpha_S$) components of the charge transfer coefficient $\alpha(T) = \alpha_H + T\alpha_S$[16–19]. This approach allows using rate equations that expand the traditional Butler-Volmer rate equation, while maintaining a single-step hypothesis. As has been pointed out again recently[36,37], the original Butler-Volmer equation (with $\alpha_H = 0.5$ and $\alpha_S = 0$) finds its origins in early attempts to understand the (inner-sphere) HER[38], but has been shown to be only strictly valid for *single-step* outer-sphere reactions[39]. The same is true for Marcus-Hush-type theories[9]. For electrocatalysis, others[40–42] and we[23] have shown that the HER activity on the Pt group metals under high mass transport conditions approximate Butler-Volmer kinetics, when accounting for the impact of the reverse rate. These results imply that Arrhenius analysis can access information of one rate-limiting transition state, e.g., of the Volmer or Tafel steps for the HER on the Pt group or sp-metals. However, these conditions are not broadly applicable.

In text we refer to a "Butler-Volmer" regime if the bias-dependent kinetics show an approximately constant $\log A^\eta$. We chose this distinction not to diminish the great utility of the more generalized free energy framework[43], but to highlight that a large part of electrocatalyst literature typically (over)analyzes linear Tafel slopes at a constant temperature, i.e., assuming $\alpha_S = 0$. Similarly, it is broadly believed that the charge transfer coefficients $\alpha_H(\eta)$ $\alpha_S(\eta)$ should be bias-independent, i.e. that the Tafel slopes must be linear, as bias-dependent charge transfer coefficients can also $\alpha_H(\eta)$ $\alpha_S(\eta)$ arise from non-kinetic mass transport effects[44,45]. However, by disregarding bias dependent charge transfer coefficients and restricting the analysis to linear Tafel fragments, the multi-step nature of electrocatalysis is obscured.

Previously, it has been shown that bias-dependent chemical potential changes for the OER[46] can lead to non-linear Tafel slopes. Recently, we unveiled an important entropic and enthalpic impact of electric fields across the double layer and have obtained bias-dependent charge transfer coefficients that are a reflection of different kinetic regimes with bias[20–23]. Here, we show that the traditional Butler-Volmer equation and related Tafel analysis at a single temperature cannot be used to capture the complex multi-step ORR sequence, which is strongly impacted by a bias-dependent coverage, electric field effects and potentially even surface restructuring at the water-solid interface. In any case, the reaction is not limited by one rate-limiting step across potentials. In fact, temperature-dependent ORR rates were previously studied with RDEs at low current densities on Pt[14] and Ru/C-Nafion[15] in concentrated phosphoric acid (25 - 250 °C) and in 0.5 M $H_2SO_4$ (25 – 75 °C), respectively. Both ORR studies hypothesized that the rates are accelerated by a potential-dependent activation entropy, while others explored bias-dependent coverage effects[10,11]. However, neither these nor our own study[20] resolved a turning point in the kinetics and the nature of the bias-dependent charge transfer coefficients.

We rely on high MEA mass transport[42,47,48] and observe continuously changing Tafel slopes that reach a minimum in the compensation region and then increase in a region that is still dominated by the ORR kinetics, before mass transport eventually changes the slopes further (Supplementary Fig. 21). The impact of mass transport is also clearly evident for the higher loading in Fig. 1d (red), where activation parameters start to deviate from the "Butler-Volmer" region and approach an iso-current line at higher overpotentials. For the low loading, we observe a clear impact of mass transport for overpotentials > 0.425 V in the kinetic maps (Supplementary Fig. 13). Finally, we intentionally reduced the $O_2$ concentration in the feed and observe that this does not appreciably change the compensation at low current densities, but primarily suppresses the pre-exponential factor at higher bias (Supplementary Fig. 14). This might explain why previous RDE studies did not observe the turning potential, as they were limited by the $O_2$ solubility in liquids. More discussion in Supplementary Note 1.

Previously, we observed very similar compensation slopes across polarizable interfaces and correlated the activation energy and pre-exponential factor with excess charge and bias-dependent space charge capacitance[22] for isolated solvation kinetics in bipolar membranes and observed compensation for reactions with theoretically predicted charged intermediates[20,21], that we also tracked with *operando* spectroscopy[21]. In fact, we correlated the increasing activation energy and pre-exponential factor in the compensation region for the OER with increasing water ordering, as observed via second harmonic generation by Speelman et al.[49,50]. Similarly, Hall and coworkers[51] observed a bias-dependent water response for the CO reduction on Cu surfaces via Raman spectroscopy in a region of increasing pre-exponential factors that partially-compensated with the activation energy[52]. In general, when timescales of ionic motion are coupled with those of the environment, electric-field-dependent changes the hydrogen bond network[53] could impact both the activation entropy and enthalpy, due to their fundamental link via the partition functions of the initial and/or transition state.

The build-up of excess charge at catalyst surfaces that are operated at potentials > ~ 0.2 $V_{RHE}$ is intimately linked to (pseudo-capacitive) electrosorption and oxidation state changes. Previously[20,21], we argued that pseudo-capacitive processes, e.g., $Pt - OH + e^- + H^+ \rightarrow Pt + H_2O$ might quench some of the excess charge *via* the formation of charge-compensated chemical bonds, in contrast to a polarizable metallic surface. As a result, a higher overpotential is needed to either stabilize the excess charge or change the surface chemistry altogether, e.g., into a metallic state. The importance of uncompensated charge in modifying the double layer and related adsorption processes is also consistent with previous observations[54]. For example, the ORR kinetics on the Pt(111) surface[55] exhibit a pronounced pH dependence, indicative of an interplay between interfacial excess charge and water dipole structures[56,57]. Further, the reaction appears sensitive to the stabilization of a (potentially) soluble, charged intermediate[58], which depends on electrostatics within the double layer[59–61]. Combined, these studies provided some insights into the role of excess charge on ORR kinetics. However, the observations were limited to low current densities and overpotentials and, in fact, ambient pressures.

When we extract the formal activation parameters ($\log A^{RHE}, E_A^{RHE}$), obtained from Arrhenius analysis at different absolute potentials, $E_{RHE}$ (Fig. 1e) we discover that the maximum $E_A^{RHE}$ almost exactly coincides with the reduction peak of $Pt - OH$ in the voltammogram in the absence of $O_2$, irrespective of the loading (Fig. 1f). The link to the reduction peak is obscured in Fig. 1d, because the kinetics at a fixed overpotential ($\log A^\eta, E_A^\eta$) are extracted from temperature-dependent currents over a range of absolute potentials (vs. RHE), due to the temperature dependence of the equilibrium potential (Supplementary Note 3 and Supplementary Fig. 20). Note, the kinetic maps are extracted from steady-state Faradaic currents (Supplementary Fig. 5) and not pseudo-capacitance. This is evident in Fig. 1f, where the steady-state ORR currents, including in the compensation region, are orders of magnitude higher than the (transient) reduction peaks in the voltammograms. In fact, the initial compensation regime extends over decades of current density, >100 mA cm$^{-2}$ for the higher loading sample. These results are critical and evidence that the bias-dependent activation parameters not only for the OER[21], but also for the ORR are closely linked to pseudocapacitive changes and possibly restructuring at the dynamic catalyst-solution interface. However, in contrast to the OER, the elevated pressures and current densities in our study here challenge current operando spectroscopy[62]. Therefore, we explore other aspects of the bias-dependent activation parameters.

The turning potential between the compensation and "Butler-Volmer" region in Fig. 1d is important for the catalyst activity. For the OER on Ni(OH)$_2$[21], we observed that the turning point occurs right at a kinetically-driven phase transition between two crystal structures[46,62,63], which is closely related to a metastable active phase (frustrated phase generated during operation). This takes place in parallel, and might be linked, to the ordering in the interfacial hydrogen bond network[49,50]. Across OER catalysts, the maximum $E_A$ and $\log A$ were essentially independent of the loading and, instead, the activity was reflected in the overpotential where the maximum $E_A$ and $\log A$ were reached[21]. In Fig. 1d-e we observe that the absolute values of the activation parameters are impacted by the loading and that the loading impacts the compensation slope in a non-trivial way. At low bias, the pre-exponential factor increases with loading, whereas it decreases at higher bias, which highlights the apparent nature of the activation parameters. Despite these complexities, we observe, as for the OER[21], that the ORR overpotential of the turning point with the maximum apparent activation energy and pre-exponential factor does not depend on loading (Fig. 1d-e) or current normalization procedure (Supplementary Fig. 22-23), and is pinned to redox processes on the catalyst surface.

The identification of intrinsic activity descriptors in industrially-relevant GDEs is very important. The overpotential at a fixed current density or the poorly-defined "onset potential" are both impacted by the surface area normalization procedure and catalyst loading (Fig. 1f). As a result, it is often argued that only well-defined single crystal surfaces or shape- and size-controlled nanoparticles can lead to fundamental insights of electrocatalyst kinetics. Here, we show that detailed insights into the intrinsic catalyst activity can be gained in (lower loading) GDEs in well-conditioned MEAs. This might enable disentangling the impact of catalyst restructuring and compositional changes in industrial-relevant systems that are operated at high current densities and pressures and over extended times. At the same time, Fig. 1g also shows, that the absolute values of log A and E$_A$ depend on the loading for the ORR. Therefore, well-controlled single crystal or shape-controlled nanoparticle studies will be critical to study the nature and, in particular, statistical distribution of active sites and link it to the kinetic maps.

**Extracting reaction order and rate constant from the pressure-dependent ORR currents**

Elevated pressure operation is critical for industrial fuel cell applications. Yet, due to the prominence of rotating disk electrodes, the kinetic effects of varying reactant pressures are almost absent in fundamental research. Figure 2a shows increasing Pt/C ORR currents with increasing O$_2$ pressure at 25 °C, while the H$_2$ pressure at the HOR GDE was held constant. Therefore, to understand the impact of the O$_2$ pressure on ORR kinetics, we performed bias- and pressure-dependent Arrhenius analysis (a complete repetition of the whole experiment can be found in Supplementary Fig. 24). We observe two distinct O$_2$ pressure-dependent changes in the kinetic maps (Fig. 2b). The compensation slope $\Delta \log A \cdot \Delta E_A^{-1}$ at low bias increases with O$_2$ pressure whereas the transition between the low compensation and fast Butler-Volmer region is systematically pushed to lower activation energies and pre-exponential factors.

In general, changing the O$_2$ pressure can increase the rate, $\nu \propto [O_2]^n k_{ORR}$, due to an increasing reactant concentration, $[O_2]$, depending on the reaction order, $n$, or an impact on the rate constant, $dk_{ORR} \cdot dpO_2^{-1}(\eta)$. Thus, we first extract the bias-dependent reaction order from the pressure-dependent current (Fig. 2c). For details about the analysis and full data set, see Supplementary Note 4 and Supplementary Fig. 25. Fig. 2d compares the bias-dependent reaction order at 25 °C with changes in the Arrhenius pre-exponential factor with O$_2$ pressure, $d \log A(\eta) \cdot d^{-1} pO_2(\eta)$ that are obtained from pressure-dependent Arrhenius analysis (Fig. 2b).

We relate the overpotential-dependent reaction order to the increasing impact of O$_2$ coverage and structural surface processes beyond the initial compensation region. At low bias, we hypothesize that the kinetics are impacted by an interfacial solvation (pre-)step, where a changing reactant concentration at the surface would have a smaller effect than the overall net excess charge. In contrast, once the first rate-limiting step is fast, an increasing reactant concentration can additionally drive the rates, in addition to the bias-dependent rate constant. However, we note that, especially once the reaction order is high, an increasing O$_2$ concentration might compete with other reaction intermediates in the multi-step sequence. Therefore, the bias-dependent changes cannot solely be assigned to the impact on the first step. Irrespectively, when we compare the bias-dependent reaction order (black curve in Fig. 2d) with the bias-dependent $d \log A \cdot dpO_2^{-1}$ (red curve), we find a deviation. The decreasing and partially-negative $d \log A(\eta) \cdot d^{-1} pO_2(\eta)$ with bias cannot be fully explained by the positive and increasing reaction order. This conflicts with pictures that assign all pressure-dependent changes to the concentration and suggests an impact of the pressure on the rate constant. We address this important point later, due to the simplifications that we need to analyze this further.

The analysis of the O$_2$-pressure-dependent ORR currents enables us to not only extract the reaction order, $n$, but to isolate the ORR rate constant, $k(\eta, T, P)$, with the explicit assumption that proton and water reaction orders can be set to zero, as detailed in Supplementary Note 4. Here, we study the kinetic region that is not strongly impacted by water or proton mass transport (Supplementary Notes 1-2), i.e., these terms remain approximately constant in our MEAs for current densities $\leq 250$ mA cm$^{-2}$. Note, $k(\eta, T, P)$ might still contain coverage terms of reaction intermediates of the multi-step sequence that only form after consuming O$_2$, next to entropic and enthalpic effects of rate-limiting transition steps.

Figure 2e shows the overpotential-dependent $k$ at different temperatures, mirroring the relative changes of the overpotential-dependent ORR currents in Fig. 2a and highlighting the very nature of electrocatalysis. Note, the dependence of the rate constant on the electrochemical potential gradients of the electrons is sometimes neglected in the literature when a special chemical nature of an intermediate is proposed. This might quickly lead to physically wrong pictures, such as when it is proposed that multiple electrons are transferred simultaneously to understand increasing currents with a supposedly unchanging rate constant[64,65]. Irrespectively, the temperature-dependence in Fig. 2e can also be used to extract a kinetic map. As shown in Fig. 2f and Supplementary Fig. 25, after accounting for the concentration term and reaction order in the Arrhenius pre-exponential factor, this kinetic map remains very similar to the maps in Figs. 1d and 2b with an almost identical turning potential. This might further imply entropic and enthalpic changes in a rate-limiting transition state, instead of a purely concentration-driven rate increase.

We also discover a pressure-independent kinetic limitation. Even though $E_A$ decreases with pressure (e.g., in Fig. 2b), the overpotential of the maximum $E_A$ barely changes. In Fig. 1, we showed that the overpotential (-0.354 V) of the maximum $E_A^\eta$ in the kinetic map corresponds approximately to the potential (0.82V$_{RHE}$) of the maximum $E_A^{RHE}$. With results in Fig. 2b, this suggests that the pressure insensitivity of the overpotential of the maximum $E_A^\eta$ is linked to a limitation related to the surface oxide structure in contact with water. Thus, we want to understand how different surface chemistry gives rise to different ORR kinetics and activity.

**Understanding ORR catalyst activity differences**

Fig. 3a shows the overpotential-dependent kinetic map for the ORR on Rh/C, Ru/C, Ir/C and Pt/C, extracted from Arrhenius analysis reaching very high linear regression R$^2$ values (up to 0.999) (Supplementary Figs. 15–18). In contrast to the overpotential-dependent activation

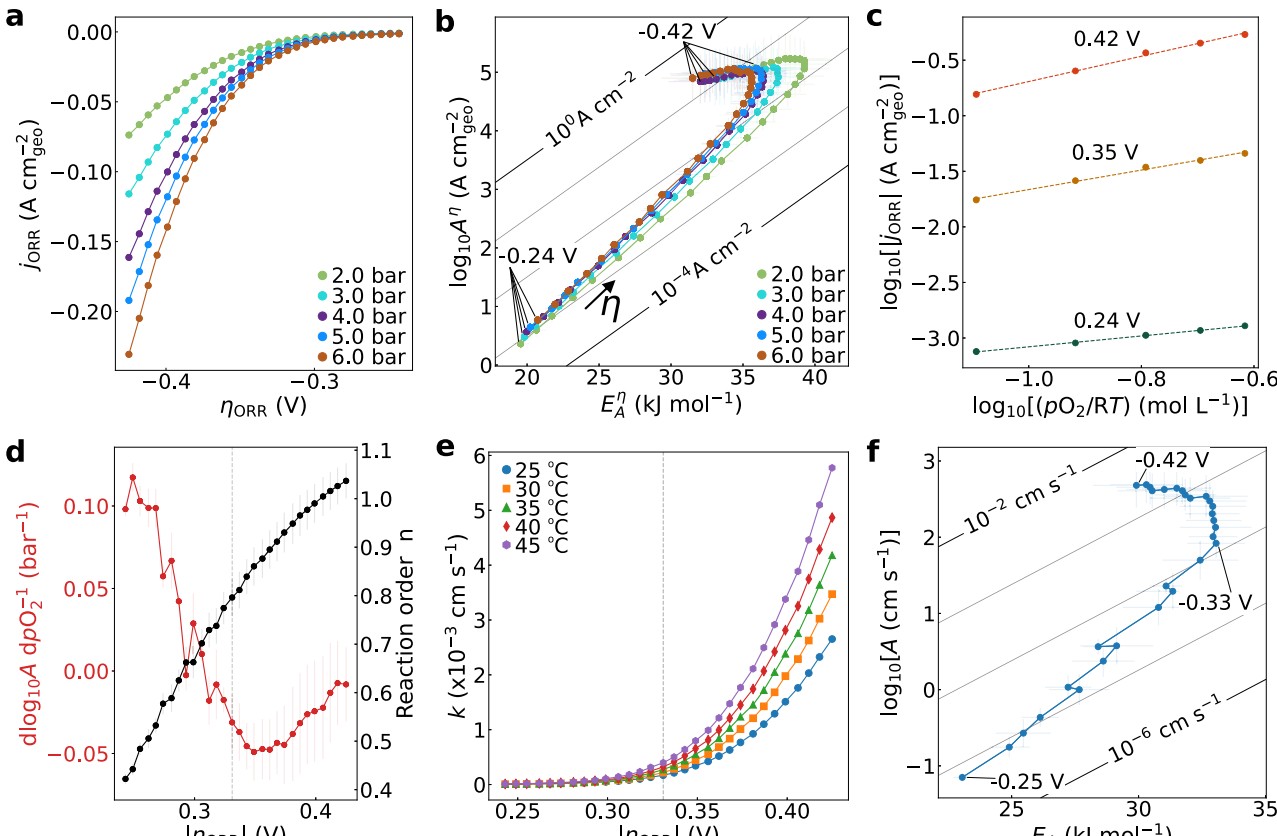

**Fig. 2 | Analysis of pressure-dependent oxygen reduction reaction (ORR) currents to extract reaction order and ORR rate constant. a** ORR polarization curves at 25 °C and different oxygen pressures $pO_2$. **b** Overpotential-dependent kinetic maps ($\log A^\eta vs. E_A^\eta$) for different pressures. The arrow in green indicates the direction of increasingly negative overpotentials. The diagonal lines are iso-current lines calculated at 25 °C. Increasing $pO_2$ primarily increases the slope in the initial compensation region and decreases the maximum $E_A^\eta$ and $\log A^\eta$ of the transition point in the kinetics. **c** Log-log plot of overpotential-dependent pressure dependence of the ORR currents at 25 °C, which provide the overpotential-dependent reaction order, $n$. For details of the analysis see Supplementary Note 4 and Supplementary Fig. 25. **d** Overpotential- dependent reaction order, $n$, at 25 °C, extracted via the approach shown in panel c and pressure dependence of the Arrhenius pre-exponential factor, $d\log A(\eta) \cdot d^{-1}pO_2(\eta)$, extracted from (**b**). **e** Overpotential-dependent rate constant, after correcting for $O_2$ concentration and reaction order and assuming a bias- and temperature-independent proton and water concentration, as detailed in main and Supplementary Notes 1, 2 and 4. **f**, Kinetic map extracted from (**e**), mirroring the kinetic maps of Fig. 1. For the $O_2$-pressure-dependent studies, the $H_2$ pressure of the Nafion-separated reference electrode is constant. For b, $\log A^\eta$ and $E_A^\eta$ are obtained from Arrhenius analysis, based on five temperatures. Error bars are based on linear regression standard error. For all studies, a Pt mass-loading of 190 μg cm$^{-2}$ was used. Overpotentials are corrected for the temperature- and pressure-dependence of the ORR equilibrium potential, the ohmic (*IR*) drop and HOR overpotential of the counter electrode. All measurements were conducted in Nafion-based membrane electrode assemblies.

parameters for Pt/C, many more distinct changes are apparent. Whereas Ir/C also displays a pronounced compensation region at low overpotentials, Ru/C and Rh/C begin in a regime with decreasing $E_A^\eta$. The overpotential ranges are different since we cannot reliably study very low or much higher current densities with our current MEA setup. Irrespectively, even for Ru/C and Rh/C apparent $\log A^\eta - E_A^\eta$ compensation appears at higher overpotentials, which could be caused by the emergence of charged intermediates and related double layer effects[21] or intricate coverage effects for multistep reactions[34].

Figure 3b shows the overpotential dependence and linkage of $E_A^\eta$ and $\log A^\eta$ in units of cm s$^{-1}$, i.e., after accounting for the $O_2$ concentration in the Arrhenius pre-exponential factor, as discussed earlier. Starting at low overpotentials, both, Pt/C and Ir/C show a distinct $\log A - E_A$ compensation where the maximum values are reached at very similar overpotentials ~350–400 mV. Further, even though the current densities are too low to accurately sample the low overpotentials for Rh/C, the initial $\log A$ and $E_A$ are also maximized around ~350–400 mV. Noteworthy, the overpotentials of the compensation region and the maximum (apparent) $E_A$ for Pt/C, Rh/C and Ir/C span a range of ~ 300–400 mV and 25–45 kJ mol$^{-1}$, respectively, and are very similar to the ones for the OER[21], ~ 200–300 mV and 20–40 kJ mol$^{-1}$,

respectively. For the OER, we were able to link the activation parameters for Ni(OH)$_2$ to structural and oxidation state phase transitions and interfacial water ordering observed with laser spectroscopy by others[49,50]. These findings further indicate[20–23] that the initial compensation and polarization of the ORR catalyst surface at lower overpotentials might be related to the build-up of excess charge and electric-field-dependent double layer effects. On the other hand, the increasing availability of active sites due to the reductive removal of adsorbed hydroxyls and oxide species has been proposed previously to understand high ORR activity on Pt$_3$Ni single crystal surfaces[11]. Whereas such bias-dependent coverage effect could also, in principle, explain increasing apparent activation energies[34], it is currently not clear how sensitive such effects are to the exact conditions, including structural changes on the highly dynamic catalyst surface. To answer such questions, *operando* spectroscopy in conjunction with electrochemical Arrhenius analysis and microkinetic modelling will be critical.

When we compare the results of Fig. 3b to Fig. 3a, we see that the correction for the reaction order and concentration term in the prefactor has the largest impact for the low activity Rh/C and Ru/C and leads to a suppression of the bias-dependent changes in $\log A$ (cm s$^{-1}$), compared to $\log A$ (A cm$^2$) at high bias. In contrast, the large bias-

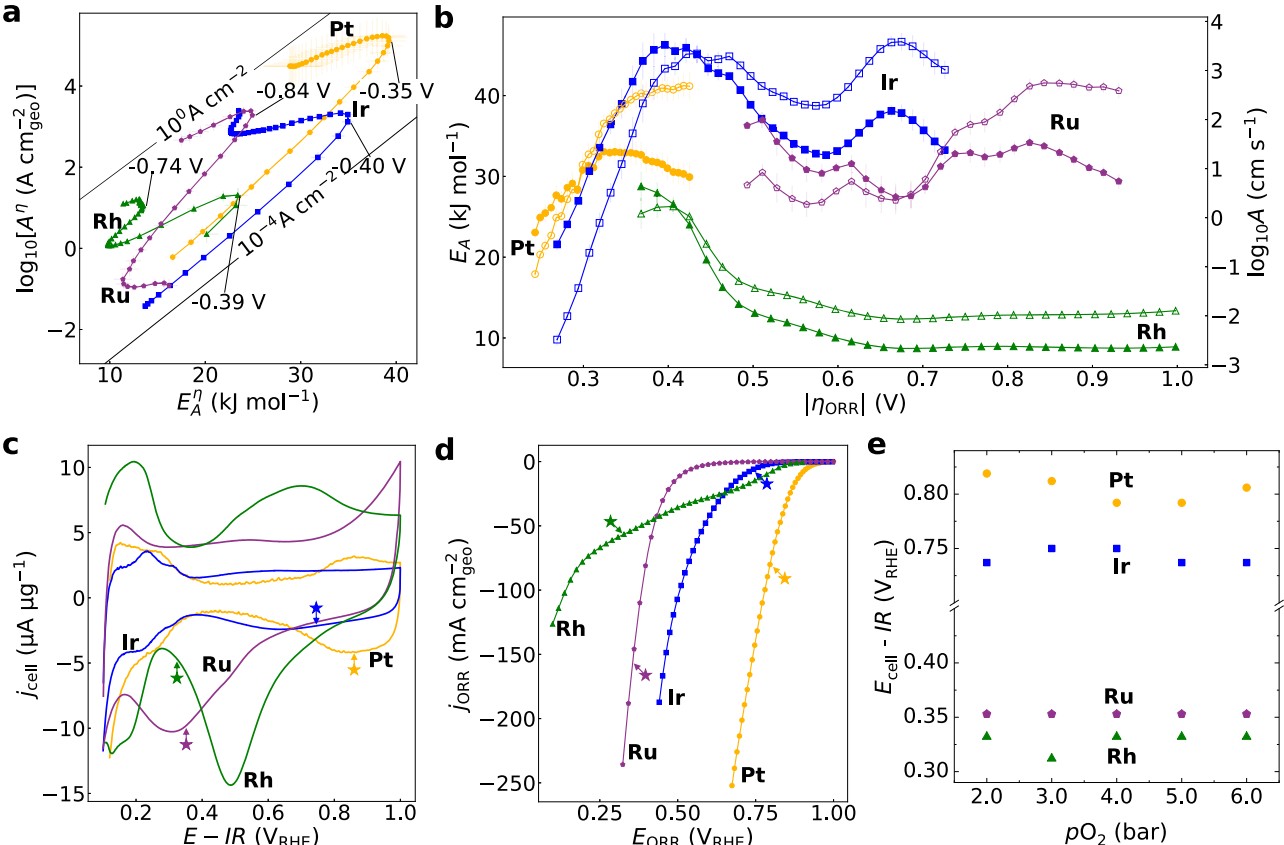

**Fig. 3 | Electrochemical Arrhenius analysis of oxygen reduction reaction across different nanoparticles in Nafion-based membrane electrode assembly.** The rate limiting steps change their degree of rate control as a function of bias. **a**, Overpotential-dependent apparent activation energy ($E_A^\eta$) and pre-exponential factor ($\log A^\eta$) in units of geometric (geo) current density, (A $cm_{geo}^{-2}$) for the ORR on Pt/C (yellow), Ir/C (blue), Rh/C (green) and Ru/C (purple) as extracted from Arrhenius analysis at 2 bar. **b**, Overpotential-dependent $E_A^\eta$ and $\log A^\eta$ in units of cm $s^{-1}$, i.e. extracted from Arrhenius analysis at different pressures and after correcting for reaction order. The solid symbols show $E_A$ and the hollow symbols $\log A$. **c**, Cyclic voltammograms at 10 mVs⁻¹ with characteristic transient reduction peaks. **d**, Faradaic, steady state ORR polarization curves extracted from chronoamperometry at 25 °C. **e**, The potential of maximum $E_A^{RHE}$ (Supplementary Fig. 27)

barely changes with pressure, whereas the reduction of maximum $E_A^{RHE}$ dominates the increasing kinetics with pressure. Importantly, the absolute potentials appear to be pinned to and limited by pseudo-capacitive processes at the water-solid interface, as indicated by the stars in panel c and d. Pt/C is shown in yellow, Ir/C in blue, Rh/C in green and Ru/C in purple. $\log A$ and $E_A$ are obtained from Arrhenius analysis, based on five temperatures. Error bars are based on linear regression standard error values. Overpotentials are corrected for temperature- and pressure-dependence of ORR equilibrium potential and HOR overpotential. A mass loading of 190 µg cm⁻² (Pt), 250 µg cm⁻² (Ir), 190 µg cm⁻² (Rh) and 180 µg cm⁻² (Ru) was used. All measurements were conducted in Nafion-based membrane electrode assemblies.

dependent changes in the initial compensation region for Pt/C and Ir/C are largely maintained after correcting for the reaction order (see Supplementary Fig. 26). Further, (loading-normalized) cyclic voltammograms in absence of O₂ across the four catalysts are shown in Fig. 3c. We find that Ir/C shows the most extended compensation at low bias, albeit the smallest (loading-normalized) pseudo-capacitance, closely followed by Pt/C. In contrast, Rh/C and Ru/C possess large pseudo-capacitance and can be studied reliably only at higher bias. The tendency for higher ORR activity with more metallic catalysts, has been reported previously, and, as mentioned earlier, rationalized by the increased availability of free sites with decreasing coverage of adsorbed hydroxyls or surface oxides[11,66]. However, Pt/C shows higher capacitance than Ir/C, yet, is more active. Furthermore, the highest bias-dependent coverage effects should be expected for Rh/C and Ru/C that undergo large reduction waves. However, we find that the apparent compensation regions are far smaller than for Ir/C and Pt/C. We hypothesize that the more metallic surfaces might build up more excess charge with bias, leading to higher pre-exponential factors and activation energies. The stars in Fig. 3c mark potentials of the maximum $E_A^{RHE}$ in the kinetic maps extracted from temperature-dependent currents at a constant potential vs. RHE (Supplementary Fig. 27) and are clearly linked with reduction peaks (Pt, Ir, Ru) or the onset of a

reduced surface (Rh). Regardless of our solvation pre-step hypothesis, at higher bias the surface processes likely dominate the activation parameters.

The ORR activity trend (high-to-low) of Pt/C, Ir/C, Ru/C and Rh/C is shown in Fig. 3d and is neither fully dominated by $\log A^\eta$ nor $E_A^\eta$. Such behavior between catalysts was observed by others and us for the HER in acid and in base[23,67–69]. Whereas we only extract the apparent activation parameters, such behavior is difficult to reconcile with the BEP relationships that assume a constant activation entropy. In fact, our results clearly contrast findings reported in pioneering DFT electrocatalysis studies on the ORR[70]. In the past, theoretical computational studies have often assigned the origin of the overpotential and rate control to a rate-limiting intermediate binding energy on a static metallic surface and predicted volcano activity plots. Further, from the data in Fig. 3d it is currently difficult to reconcile how the elegant scaling relationships[71] between chemically-similar intermediates alone could be applied to explain the bias dependent apparent activation parameters. Even neglecting our solvation pre-step hypothesis, the bias-dependent catalyst surfaces undergo substantial chemical and structural changes, some of which leave behind clear fingerprints in the apparent activation parameters, such as the maximum activation energy at low bias.

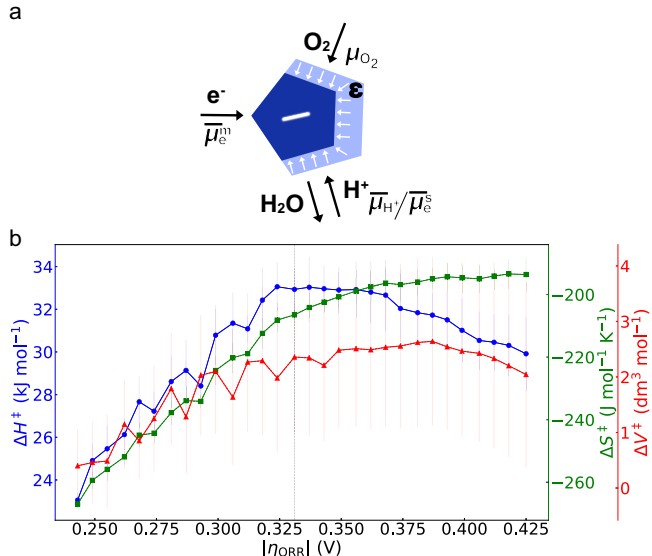

**Fig. 4 | Bias-dependent polarization at the triple-phase boundary and activation parameters for a hypothesized rate-limiting transition state.**
**a** Electrocatalysts (blue pentagon) can be exposed to a free energy driving force by applying a difference between the electrochemical potential of the electrons in the electrode, $\bar{\mu}_e^m$, and in solution $\bar{\mu}_e^s$. The latter is related to the electrochemical potential of the protons, $\bar{\mu}_{H^+}$, via the reversible hydrogen evolution reaction, $H^+ + e^- \rightarrow 1/2H_2$. Additionally, the chemical potential of the $O_2$ reactant, $\mu_{O_2}$, can be varied via the $O_2$ partial pressure. These applied free energy differences can either drive reactions over static, charge-neutral surfaces or they can lead to local chemical potential changes, i.e., build-up of excess charge and intermediate coverage changes. Excess charge leads to a change in the dielectric solvent (bright blue), $\epsilon$, which can impact the interfacial charge transfer of $H^+$ and OH⁻.
**b** Approximated, bias-dependent activation enthalpy ($\Delta H^{\ddagger}$), activation entropy ($\Delta S^{\ddagger}$), and activation volume ($\Delta V^{\ddagger}$), which are rooted in the common partition functions of the initial (i) and the transition state (‡), $Z^i(\eta, P, T)$ and $Z^{\ddagger}(\eta, P, T)$, respectively. Note, for the calculation, a range of simplifications have been made (see Supplementary Note 5). Most importantly, it is assumed that one rate-limiting transition state dominates the kinetics at low bias. Error bars are based on linear regression standard error values. Overpotentials are corrected for the temperature- and pressure- dependence of the ORR equilibrium potential. For all measurements, a Pt mass-loading of 190 μg cm⁻² was used. All measurements were conducted in Nafion-based membrane electrode assemblies.

In Fig. 3e we plot the potential $E_{RHE}$ where the maximum $E_A^{RHE}$ are reached (Supplementary Fig. 27) and observe that the potential barely changes with pressure, but appears pinned by pseudo-capacitive oxidation state changes at the water-solid interface (Fig. 3c). Thus, we conclude that in all generality, for multi-step reactions, the rate limiting steps and transition states can change their degrees of rate control[32,33] with bias. This needs to be accounted for in new theoretical work and linked to the structural and chemical changes at the bias dependent catalyst-solution interface.

The discovery of rich bias-dependent activation parameters that cascade through a range of rate-limiting steps speaks to the power of temperature-dependent kinetic studies in systems that provide high mass transport. Whereas we cannot a priori extract absolute activation parameters with high precision for multi-step reactions, we started to link the activation parameters to chemical and structural changes. With more operando spectroscopy, fundamental electrochemistry and updated theory, these kinetics can be dissected and linked to the degrees of rate control of individual transition states[32,33]. We contrast these rich insights to the bias-dependent Tafel slopes for Rh/C in Supplementary Fig. 28, which reach a minimum at 350–400 mV, before increasing to 1 V dec⁻¹. Despite these high values, the polarization curves are far from mass transport-limited, but contain kinetic information that can be revealed by studying their temperature dependence.

## Considerations on a pressure-dependent rate constant

According to statistical mechanics[8,72], the transition state is characterized not only by the activation enthalpy and entropy, but also the activation volume, which relates the susceptibility of the rate constant to a changing pressure in the reacting phase. Traditionally, for reactions in solution, the activation volume is obtained by varying the hydrostatic pressure, thereby informing on the solvent's molecular response in the transition state[73–75]. However, at the triple-phase boundary in electrocatalysis (Fig. 4a), the adsorption of gas reactants is often implicated in rate-limiting steps, such as during the $CO_2$, $N_2$ and $O_2$ reduction reactions. Thus, the pressure-dependent chemical potential changes in the transition state complex are of broad interest.

In Fig. 2d, we extracted the bias-dependent reaction order, $n$, for Pt/C, and observed that it cannot explain the changes of $d \log A(\eta) \cdot dpO_2^{-1}(\eta)$. Thus, as detailed in Supplementary Fig. 29 and Supplementary Note 4 and 5, we extract the ORR's kinetic susceptibility to pressure. We refer to this susceptibility as the activation volume, $\Delta V_{\eta}^{\ddagger} = - RTd\ln k \cdot dpO_2^{-1}(\eta)$, based on the assumption that our results inform of the partial molar volumes of the transition and reactant states of a rate-limiting step that is largely dominating the kinetics at low overpotentials. Regardless of whether our pressure-dependent data has this physical meaning, this method allows for a qualitative comparison of the effect of pressure and overpotential on kinetics. The absolute values remain inaccessible without more information about surface chemistry, solvent and coverage effects[32] for the complex multistep sequence.

Figure 4b compares the bias-dependent activation volume $\Delta V_{\eta}^{\ddagger}$ with the bias-dependent activation enthalpy, $\Delta H_{\eta}^{\ddagger}$, and entropy, $\Delta S_{\eta}^{\ddagger}$. Strikingly, all activation parameters rise and compensate until 325 mV overpotential. At higher bias, $\Delta H_{\eta}^{\ddagger}$ and $\Delta V_{\eta}^{\ddagger}$ fall together, whereas $\Delta S_{\eta}^{\ddagger}$ saturates. Noteworthy, $\Delta V_{\eta}^{\ddagger}$ is obtained from ORR currents at constant temperature but variable pressures, whereas $\Delta H_{\eta}^{\ddagger}$ and $\Delta S_{\eta}^{\ddagger}$ are obtained from ORR currents at a variable temperature and variable pressures. Further, the bias-dependent trends are also apparent in $E_A^{\eta}$ and $\log A^{\eta}$ in Figs. 1d and 2b, that are obtained from ORR currents at variable temperature and constant pressure. Thus, these bias-dependent changes in the activation parameters might have a shared root cause.

The activation parameters $\Delta S^{\ddagger}$, $\Delta H^{\ddagger}$ and $\Delta V^{\ddagger}$ are all dependent on the partition functions of the transition and reactant state, $Z^{\ddagger}$ and $Z^R$, respectively. For example, if we assume a constant $Z^R$ ($Z^{\ddagger}$), the bias-, pressure- and temperature-dependent $Z^{\ddagger}(\eta, P, T)$ ($Z^R(\eta, P, T)$) would interrelate all changes in the activation parameters and any change in $\Delta H_{\eta}^{\ddagger}$ would also be reflected in $\Delta S_{\eta}^{\ddagger}$. We speculate, that the bias- and pressure-dependent formation of a charged intermediate up to $\eta \sim 0.325V$ could explain why the bias-dependent changes in $\Delta H_{\eta}^{\ddagger}$ and $\Delta S_{\eta}^{\ddagger}$ are also reflected in $\Delta V_{\eta}^{\ddagger}$. On a molecular level, a charged superoxo $O_2^-$ (or peroxo $O_2^{2-}$) intermediate has been proposed previously for the ORR[76], i.e. $O_2 + e^- \rightarrow O_2^-$, as a rate-limiting step at low overpotentials. This would allow the $O_2$ pressure to impact the formation of such a rate-limiting charged intermediate and, thus, impact the electric-field-dependent double-layer structure. However, in general, the excess charge likely redistributes in the highly dynamic molecular environment with bias, potentially starting with charged $O_2^-$ intermediate, but then transitioning to electronic spillover effects[77] from the metal into an adsorbed $O_2$ layer or many other variations, given the highly dynamic nature of the surface, that likely undergoes substantial restructuring, especially at these elevates pressures. In fact, irrespective of the hypothesized rate-limiting transition state, the results in our study challenge the whole notion that rate-limiting intermediates do not change their nature and energetics with bias, which is also explored in new computational work[78,79]. In general, the atomic metal

arrangement and oxygen content can be changing and responding to the $O_2$ pressure.

At higher bias, we observe that the entropic barriers become quasi-potential independent, which might reflect kinetics that are controlled by adiabatic electron transfer, where the bias primarily impacts $\Delta H_\eta^\ddagger$, because the electron moves on timescales that are much faster than the ones of the nuclear motions in the environment (Born-Oppenheimer approximation). However, we reiterate that the changes in the activation parameters in Fig. 4 are likely greatly impacted by complex chemical and structural changes that could evade the simple analysis performed above. Therefore, a higher pressure range will be needed to test our hypothesis in the future.

## Discussion

By performing temperature-dependent electrochemistry and Arrhenius analysis under high mass transport conditions, we are able to obtain pressure- and bias-dependent kinetics that lead to a substantially updated picture of the ORR, and electrocatalytic multi-step reactions in general. Collectively with our recent results[20–23], we identify again an initial compensation region at low overpotentials, which is likely related to the change of the local chemical potentials, encompassing a bias-dependent intermediate coverage and electric field effects inside the double layer, all of which can lead to different degrees of rate control of different rate limiting steps and transition states. Further, we observe strong similarities with our findings for the OER[21], where the initial compensation region extends across a similar range of ~350–400 mV, strongly implying shared kinetic limitations at the catalyst-solution interface.

At higher overpotentials, we observe kinetic changes that likely stem from complex bias-dependent intermediate coverage and structural changes that impact the apparent activation parameters and shift the rate-limiting steps[34]. These transitions have been overlooked in previous fundamental electrocatalyst studies that were limited to low current densities and overpotentials, near-equilibrium surface chemistry and often relied on analyzing linear Tafel fragments. In fact, even Conway focused on (temperature-dependent) linear Tafel slopes for sp-metals[16–19] to maintain the single-step hypothesis, whereas the bias-dependence of the charge transfer coefficient and the Tafel slopes has received even less attention. Historically, the overreliance of fundamental electrocatalysis research on rotating disk electrodes with substantially lower mass transport (that can also lead to bias dependent Tafel slopes) might have substantially slowed progress.

The applied electrochemical overpotential can increase the ORR rates much more readily than the $O_2$ pressure in the 1-6 bar range. In general, acknowledging different reactions conditions, electrochemical technology is able to operate at much lower pressures than thermo-catalysis. For example, $CO_2$ hydrogenation requires much higher pressures ( > 30 bar) in thermo-catalysis compared to high rate electrocatalytic $CO_2$ reduction at ambient pressures. For $CO_2$ reduction, the formation of a charged intermediate is thought to be critical, which can be readily generated under bias and stabilized at the interphase between solid catalyst and the electric-field dependent hydrogen bond network. Therefore, our results and hypothesis around the bias- and pressure-induced pre-organization and polarization[20–23] of the catalyst-solution interface might help explain the superior role of the electrochemical bias in pre-organizing transition states in catalysis.

## Methods
### Electrodes
Electrodes were fabricated by spray-coating inks made from commercial dispersions of nanoparticles onto heated porous carbon paper substrates (Freudenberg, H23C2). Carbon paper sheets measuring $5.3 \times 5.3\ cm^2$ were used for the coating process, and smaller $1 \times 1$ $cm^2$ pieces were cut for use as electrodes in the MEA cell. Detailed

information on the properties of the commercial nanoparticle dispersion is provided in Supplementary Table 1.

### Ink Preparation
Isopropanol-based inks were prepared by weighing the components into 10 mL borosilicate-glass vials. The nanoparticle dispersion, used as the base material, was mixed with Nafion ionomer. Homogenization was performed using a horn sonicator immersed in the ink, with the vials placed in an ice bath to prevent overheating and thermal aggregation.

For standard inks, the sonication protocol consisted of 5 s active pulses alternated with 3 s inactive intervals, totaling 15 minutes of active sonication at 60% power. In the case of Pt-black-based inks, the higher Pt loading necessitated more extensive homogenization, requiring 45 min of active sonication at 70% power to ensure stability and uniform dispersion. A comprehensive table detailing the precise masses of each component used in ink preparation, as well as the coating procedure, can be found in Supplementary Table 2.

### Cell Assembly
The MEA cell was constructed using graphite blocks with a serpentine-patterned square flow field measuring $2.1 \times 2.1\ cm^2$ (Supplementary Fig. 30). A Nafion 212 membrane ($2.2 \times 2.2\ cm^2$) was pre-hydrated for at least 24 h before assembly. The membrane was positioned between two $1\ cm^2$ GDL electrodes embedded in 200 μm-thick Teflon gaskets. MEA hardware screws were tightened to a torque of 4 Nm to ensure proper sealing and contact.

To optimize pressure distribution, a stainless-steel compression jacket with a raised $3.0 \times 3.0\ cm^2$ central area was placed over the flow field. Pressure-sensitive paper was used to evaluate and confirm the uniformity of pressure distribution both with and without the jacket. Further experimental details and the pressure distribution are available in Supplementary Fig. 30.

### Experimental Setup
The gas supply consisted of two independent lines regulated by mass flow controllers (MFCs). Each line is connected to a humidification bottle for adjusting the relative humidity of the supplied gases, with a bypass pathway allowing the direct delivery of dry gases. One line supplied pure $H_2$, while the other, equipped with a manifold containing multiple MFCs, facilitated the delivery of various gases. A downstream gas mixer ensured homogeneity when mixing gases. For the ORR experiments, pure $O_2$ was used, while pure $N_2$ served as the working gas for blank measurements. See Supplementary Fig. 31 for a picture of the setup.

After passing through the MEA, the gas lines were connected to back-pressure regulators equipped with electronic valves and integrated pressure sensors operating in a negative feedback loop. Argon was used as the regulating gas to provide 10 bar input pressure to the electronic valves. These valves modulated flow to maintain equilibrium between the input and measured pressures, ensuring stability. The MEA cell was heated using metallic bars embedded in its brass blocks, with temperature electronically regulated via a K-type thermocouple and a custom-built Eurotherm 3508-based controller. A similar system controlled the temperature of the humidification bottles, outlet lines, and bypass. An image and extended description of the setup can be found in Supplementary Fig. 31. For all measurements a Gamry 3000 potentiostat was used.

### MEA conditioning and pressure control validation
To condition the MEA, the cell was heated to 55 °C and subjected to a chronopotentiometry (CP) run at -100 mA cm⁻². This step was critical for conditioning all layers of the MEA, particularly because assembly with a humidified membrane can initially lead to electrode flooding. Following this, cyclic voltammetry (CV) was performed to activate the

electrochemical interfaces, continuing until subsequent cycles became nearly invariant. The results of the MEA conditioning, along with data confirming stability of the electrochemical signals and pressure regulation, are shown in Supplementary Figs. 4–5.

To prepare the MEA for chronoamperometric (CA) measurements, the following activation protocol was implemented before each measurement: *i)* Six cycles of CV were performed within the potential limits of the chronoamperometries for activation. *ii)* A 10 min CA was conducted at the initial potential to stabilize the electrode interfaces. *iii)* Electrochemical impedance spectroscopy (EIS) was conducted at the same initial potential to determine the cell resistance.

### Chronoamperometry
For Pt experiments, the initial potential was set to 1.0 $V_{RHE}$, and CAs were conducted by sequentially decreasing the absolute potential (increasing the overpotential) in 10 mV steps across 41 repetitions, concluding at a final potential of 0.6 $V_{RHE}$. For Rh experiments, the initial potential was lowered to 0.92 $V_{RHE}$, with a potential step size of 20 mV applied over 43 repetitions, reaching a final potential of 0.06 $V_{RHE}$. For Ir experiments the initial potential was set to 1.0 $V_{RHE}$ with potential step size of 15 mV for 41 repetitions, reaching 0.4 $V_{RHE}$ as final potential. For Ru, the potential was initially set to 1.0 $V_{RHE}$ and a step size of 21 mV applied over 41 repetitions reaching 0.16 $V_{RHE}$ as final potential. Each CA maintained the applied potential for 60 s, recording the current every second. The steady-state current at each potential was determined by averaging the final 15 data points from each run.

### Temperature variation
After the conditioning at higher temperature, the first temperature was always 25 °C and then progressively increased in 5 °C steps.

### Oxygen pressure variation
following the completion of experiments within the temperature range (25 °C–45 °C), the oxygen pressure was increased by 1 bar and the temperature reset to 25 °C. The system was allowed to stabilize overnight to ensure proper equilibration before subsequent measurements.

### Data processing and interpolation
The analysis routine detailed below is schematically shown in Supplementary Fig. 19. First, current was normalized by geometric area. Next, temperature-dependent HOR crossover current was estimated for all metals by performing CV with $N_2$ instead of $O_2$ in the working electrode chamber (Supplementary Fig. 3). These reference measurements were run at each experimental pressure and temperature. The positive current displacement of the CV midline from 0 A was taken as steady state HOR current, which was normalized by geometric area and subtracted from all currents for a given pressure and temperature. Note, the HOR current (~1 mA cm⁻²) was low compared to currents used to perform the Arrhenius analysis (1-300 mA cm⁻²), including in the extended compensation region.

Uncompensated series resistance was determined from potentiostatic electrochemical impedance spectroscopy (EIS) at the lowest experimental potential for each metal (1 V for Pt and 0.9 V for Rh) for each temperature at a given pressure. Ohmic drop correction was then performed. Anodic HOR was measured at each experimental temperature, finding highly linear I-V polarization curves (Supplementary Fig. 1). The corresponding anodic HOR potential for each respective ORR current density was subtracted from the IR-corrected cell potential to produce the corrected ORR potentials vs RHE ($E_{ORR}$).

The ORR equilibrium potential ($E^0(T,P)$) was calculated for each combination of temperature and pressure. First, the reported temperature sensitivity[9] of the reaction was used to calculate $E^0$ at each

temperature from the reference value of 1.23 $V_{RHE}$ at 298 K and 1 atm (Eq. 1). These values were used in the Nernst equation (Eq. 2) to calculate $E^0(T,P)$ for each pressure at a given temperature. Note, the Nernst equation itself does not inform on the temperature sensitivity of the equilibrium potential. Next, overpotentials were calculated by subtracting the corrected $E^0(T,P)$ from IR-corrected ORR potentials.

$$E^0(T, 1\,\text{atm}) = E^0(298\,\text{K}, 1\,\text{atm}) + \frac{dE}{dT}(T - 298\,K) \qquad (1)$$

$$E^0(T,P) = E^0(T, 1\,\text{atm}) - \frac{RT}{nF} ln\frac{1\,\text{atm}}{P} \qquad (2)$$

A common set of evenly-spaced overpotentials was created using linear interpolation to compare data across pressures and temperatures. Corrected ORR current densities were interpolated from the averaged steady state data to match the *IR-* and $E^0$-corrected overpotentials.

As discussed in Supplementary Note 4, temperature- and overpotential-dependent reaction order and rate constants were calculated. Arrhenius analysis of the rate constant was performed to produce bias-dependent $E_A$ and log $A$. The enthalpic and entropic activation barriers ($\Delta H^{\ddagger}$ and $\Delta S^{\ddagger}$) were calculated from the Arrhenius analysis via the Eyring equation, as described in Supplementary Note 5. Finally, the bias-dependent activation volume was calculated from its definition in statistical mechanics using $\left(\frac{\partial lnk}{\partial P}\right)_{T,\eta}$ from the derivative rate law equation explained in Supplementary Note 5.

## Data availability
The authors declare that the source data supporting the findings of this study are available within the paper and its Supplementary Information files. Should any additional data files be needed they can be requested from the corresponding author, if it does not conflict with follow-up projects and further in-depth analysis. Source data are provided with this paper.

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

## Acknowledgements

We sincerely thank D. Teschner, T. Jones and M. Lizée for discussions around the impact of a bias-dependent intermediate coverage. ARSO and JMG acknowledge support from the German Federal Ministry for Research, Technology and Space (BMFTR) under grant no. 03SF0662B (ReveAL). JD and SZO acknowledge funding from the European Union's Horizon (ERC, ORION, 101077895). BRC and EO acknowledge funding from the German Federal Ministry of Education and Research (BMBF) under grant no. 03EW0015B (Catlab). We acknowledge open access funding provided by Max Planck Society.

## Author contributions

S.Z.O. conceived of the project. A.R.S.O. performed all experiments. J.D. developed the scripts for the analysis and plotting of the data. J.M.G. assisted with some measurements. E.O. performed the electron microscopy. A.R.S.O., J.D., and S.Z.O. analyzed the data and discussed them with B.R.C. A.R.S.O., JD and S.Z.O. wrote the manuscript with help of B.R.C.

## Funding

## Competing interests

The authors declare no competing interests.
