## [Transparent Peer Review file · Nature Communications]

Pressure and Bias Dependence of the Rate-Limiting Steps of the Oxygen Reduction Reaction

Corresponding Author: Dr Sebastian Oener

Version 0:

Reviewer comments:

Reviewer #1

(Remarks to the Author)

The manuscript by Oener and team is a provocative and highly interesting contribution to our understanding of fundamental interfacial electrochemistry and catalysis. It covers much space intellectually and the data, while highly interesting, is probably not possible to fully understand at this point. Regardless, this is a valuable contribution.

Regarding the revisions, the reviewer appreciates and commends the author's extensive response to the previous comments. Overall, the manuscript is significantly improved compared to the initial manuscript. Key additions to the manuscript include more discussion and experimental results regarding possible mass transport limitations, an effective rearranging of figures from the SI to the main manuscript, the addition of two additional catalyst materials (Ir/C and Ru/C), and more explicitly state assumptions being made throughout the main text. Overall, the manuscript reads more clearly and what is known, and what is hypothesized is now more discussed cleanly.

There remain additional issues that should be considered prior to possible publication in Nature Communications.

Remarks

1. Regarding the mass-transport effects, the j-E curves clearly show some impact of mass transport when the O₂ concentration was reduced to 20%, granted, this was a very aggressive reduction in O₂ concentration. Similarly, there are changes in the kinetic map (Figure S14a-c). It is interesting that despite clear mass transport limitations the overall shape in the kinetic map remains similar with largely the only changes being the absolute values of the kinetic parameters. The authors note that the turning point potential moves to larger overpotentials as well. In the context of their argument, this would be problematic because the turning point is argued to be related to pseudocapacitive processes. However, looking at Figure S14c it appears the turning point label is only different by 10 mV. Given the arbitrariness in describing exactly where the turning point occurs, is this not negligible and could this be due to variation in the experiments? Further clarification here would be valuable, especially given that the turning point potential is a key result put forth in this work.

2. In addition to point 1 above, the authors cite Figure 1f as further validation that mass transport is not limiting. Here, they show the rate of catalysis scales with catalyst loading which they argue is evidence that mass transport limitations do not exist. This is not necessarily true. If the system is mass transport limited via O₂ diffusion to an active site, increasing the number of active sites would indeed decrease mass transport limitations. For example, if you double the number of active sites each site supports half the current, and therefore mass transport to the site does not need to be as facile. It is also possible for mass transport to be limiting through the bulk of the carbon support (e.g., transport layer) which would cause loading-independent transport limitations but at these currents it is typically thought that it is transport to a site that is limiting.

3. The reviewer appreciates further clarification regarding the role of pressure-induced changes in the transition-state volume. The physical picture here, however, is still not entirely clear. The authors invoke pressure-effects on O₂ + e⁻ O₂⁻ involving solvation kinetics, excess charge and double layer ordering which is all unclear.

4. The authors invoke multiple times that the previous theories are for outer sphere type reactions, but this is not rigorously true. The Butler-Volmer equation was not originally derived for an outer sphere reaction and should be considered more a phenomenological expression. I encourage the authors to be careful with their language here and be precise.

5. The authors claim that their results indicate that the reactant pressure electrochemically polarizes the transition state of a rate-limiting step in a manner analogous to applied overpotential. This is confusing as stated. I suppose the authors may be describing a pressure dependent effect on the structure of the electrochemical double layer which changes effective polarization, but it's quite speculative and unclear, especially at this point in the paper and probably deserves revisions as also described above in the discussion section

6. The section titled Discussion is just a discussion of the pressured effects primarily. There's discussion and speculation mixed in across the entire paper, so distinctly labeling this discussion is not appropriate. Please provide a more descriptively accurate section title.

7. The authors write that the lack of substantial nonlinear Arrhenius behavior precludes the impact of temperature-dependent intermediate coverages or structural changes. I would be less definitive with the language in this sentence. It seems very difficult to rule out all those things.

8. The authors invoke the language of a frustrated phase transition, which has a very precise meaning in physics community. It's not clear the exact meaning they have in this study. I'm not sure it's appropriate. This should be thoughtfully evaluated to use precise language and definitions.

9. The authors include a highly critical statement regarding the use of DFT to calculate intermediate binding energies on metallic surfaces and predict rate determining steps. While this criticism is warranted due to the simplicity of those models, I would encourage the authors to be more matter-of-fact about their statements. For example, they state that one of the calculations was the most popular DFT studies ever performed. More precisely, it is one of the highest cited papers in the field of computational electrocatalysis. They also write that the authors proclaimed loudly that they have found the origin of the overpotential. However, that's also not a factual statement that belongs in a scientific paper. One can simply state that the authors and many others conclude that binding energies are primarily responsible for rate control through this analysis, but that that is inconsistent with the study here and indeed much of detailed experimental electrochemistry.

10. Overall in the paper there are many typos and grammatical errors. Please carefully revise for the final submitted version.

Reviewer #2

(Remarks to the Author)

The reviewer thanks the authors for their thorough and transparent response. The reviewer recognizes that performing kinetic analyses of GDE-based catalysts under operating conditions is highly challenging, yet valuable. The experimental data and the comprehensive analysis presented by the authors will provide important insights and serve as a meaningful reference for the electrocatalysis community. I recommend publication in Nature Communications.

Version 1:

Reviewer comments:

Reviewer #1

(Remarks to the Author)

The authors have suitably revised the manuscript, which is now ready for publication in Nature Comms.

We sincerely thank both Reviewers for carefully reading our manuscript, the positive evaluation and for the remaining detailed comments.

Reviewer 2 recommends publication in Nature Communication, while Reviewer 1 has remaining comments and question that are given below in *blue italic* font and our specific response to each comment is given in black font. Changes/additions to the manuscript text are given in **highlighted text**.

Reviewer #1 (Remarks to the Author):

The manuscript by Oener and team is a provocative and highly interesting contribution to our understanding of fundamental interfacial electrochemistry and catalysis. It covers much space intellectually and the data, while highly interesting, is probably not possible to fully understand at this point. Regardless, this is a valuable contribution.

Regarding the revisions, the reviewer appreciates and commends the author's extensive response to the previous comments. Overall, the manuscript is significantly improved compared to the initial manuscript. Key additions to the manuscript include more discussion and experimental results regarding possible mass transport limitations, an effective rearranging of figures from the SI to the main manuscript, the addition of two additional catalyst materials (Ir/C and Ru/C), and more explicitly state assumptions being made throughout the main text. Overall, the manuscript reads more clearly and what is known, and what is hypothesized is now more discussed cleanly.

There remain additional issues that should be considered prior to possible publication in Nature Communications.

Remarks

1. Regarding the mass-transport effects, the j-E curves clearly show some impact of mass transport when the O₂ concentration was reduced to 20%, granted, this was a very aggressive reduction in O₂ concentration. Similarly, there are changes in the kinetic map (Figure S14a-c). It is interesting that despite clear mass transport limitations the overall shape in the kinetic map remains similar with largely the only changes being the absolute values of the kinetic parameters. The authors note that the turning point potential moves to larger overpotentials as well. In the context of their argument, this would be problematic because the turning point is argued to be related to pseudocapacitive processes. However, looking at Figure S14c it appears the turning point label is only different by 10 mV. Given the arbitrariness in describing exactly where the turning point occurs, is this not negligible and could this be due to variation in the experiments? Further clarification here would be valuable, especially given that the turning point potential is a key result put forth in this work.

Answer: We thank the reviewer for all the time dedicated to the evaluation of our work and for the attention to the detail. We fully agree that a 10 mV shift is insufficient to make any definite claims. Therefore, the main effect is clearly the suppression of the pre-factor at higher bias. We have edited the text accordingly to avoid misunderstandings.

Page 6: Finally, we intentionally reduced the O₂ concentration in the feed and observe that this does not appreciably change the compensation at low current densities, but primarily suppresses the pre-exponential factor at higher bias (Supplementary Figure 14).

Supplementary Note 1

However, for the O₂-poor conditions, the turning point occurs at slightly higher overpotentials and the absolute pre-exponential factor and activation before the turning point are slightly increased. However, after the turning point, the mass transport limitation might result in the substantial reduction of the pre-exponential factor.

2. In addition to point 1 above, the authors cite Figure 1f as further validation that mass transport is not limiting. Here, they show the rate of catalysis scales with catalyst loading which they argue is evidence that mass transport limitations do not exist. This is not necessarily true. If the system is mass transport limited via O₂ diffusion to an active site, increasing the number of active sites would indeed decrease mass transport limitations. For example, if you double the number of active sites each site supports half the current, and therefore mass transport to the site does not need to be as facile. It is also possible for mass transport to be limiting through the bulk of the carbon support (e.g., transport layer) which would cause loading-independent transport limitations but at these currents it is typically thought that it is transport to a site that is limiting.

Answer: We thank the reviewer for this detailed explanation and agree that we cannot use this as argument to exclude mass transport limitations, which would require more analysis, including ideally being able to know through operando spectro-microscopy studies the number, density and dynamic evolution of the active sites as well as their possible kinetically driven deactivation. We believe that our experiments now included in the SI address the issue of mass transport limitations and thus, following the reviewer's comment we have decided to remove the previous statement from the text.

Page 8: For the ORR, Figure 1f shows that the current scales with catalyst loading (Fig. 1f), as expected for a kinetically controlled regime that is not dominated by mass transport effects.

3. The reviewer appreciates further clarification regarding the role of pressure-induced changes in the transition-state volume. The physical picture here, however, is still not entirely clear. The authors invoke pressure-effects on O₂ + e⁻ → O₂⁻ involving solvation kinetics, excess charge and double layer ordering which is all unclear.

Answer: We thank the reviewer for asking for more clarification. We agree that overall, this section is speculative. We have tried to clarify the hypothesis, but also highlighted the uncertainty.

Page 14-15:

We speculate, that the bias- and pressure-dependent formation of a charged intermediate up to $\eta \sim 0.325\text{V}$ could explain why the bias-dependent changes in $\Delta H_{\eta}^{\ddagger}$ and $\Delta S_{\eta}^{\ddagger}$ are also reflected in $\Delta V_{\eta}^{\ddagger}$. On a molecular level, a charged superoxo O₂⁻ (or peroxo O₂²⁻) intermediate has been proposed previously for the ORR⁷⁶, i.e. O₂ + e⁻ → O₂⁻, as a rate-limiting step at low overpotentials. This would allow the O₂ pressure to impact the formation of such a rate-limiting charged intermediate and, thus, impact the electric-field-dependent double-layer structure.

However, in general, the excess charge likely redistributes in the highly dynamic molecular environment with bias, potentially starting with charged O_2^- intermediate, but then transitioning to electronic spillover effects⁷⁷ from the metal into an adsorbed O_2 layer or many other variations, given the highly dynamic nature of the surface, that likely undergoes substantial restructuring, especially at these elevated pressures. In fact, irrespective of the hypothesized rate-limiting transition state, the results in our study challenge the whole notion that rate-limiting intermediates do not change their nature and energetics with bias, which is also explored in new computational work^{78,79}. In general, the atomic metal arrangement and oxygen content can be changing and responding to the O_2 pressure.

4. The authors invoke multiple times that the previous theories are for outer sphere type reactions, but this is not rigorously true. The Butler-Volmer equation was not originally derived for an outer sphere reaction and should be considered more a phenomenological expression. I encourage the authors to be careful with their language here and be precise.

Answer: We thank the reviewer for asking for more clarification and helping us to improve the manuscript further, as our intention was not to criticize the general free energy framework. We have now added more clarification as to why we chose to denote one kinetic region as “Butler-Volmer” region in the kinetic map. The main intention is to link it to the original Butler-Volmer equation used across the electrocatalyst literature. We have edited the text accordingly, as shown below.

Page 5-6: In particular, the temperature dependence of the Tafel slope ($b = \ln 10 RT / (F \alpha(\eta, T))$) has been traditionally interpreted via enthalpic ($\alpha_H(\eta)$) and entropic ($\alpha_S(\eta)$) components of the charge transfer coefficient $\alpha(T, \eta) = \alpha_H(\eta) + T\alpha_S(\eta)$ ¹⁶⁻¹⁹. This approach allows using rate equations that expand the traditional Butler-Volmer rate equation. As has been pointed out again recently^{36,37}, the original Butler-Volmer equation (with $\alpha_H = 0.5$ and $\alpha_S = 0$) finds its origins in early attempts to understand the (inner-sphere) hydrogen evolution reaction³⁸, but has been shown to be only strictly valid for *single-step* outer-sphere reactions³⁹. The same is true for Marcus-Hush-type theories⁹.

For electrocatalysis, others⁴⁰⁻⁴² and we²³ have shown that the HER activity on the Pt group metals under high mass transport conditions approximate Butler-Volmer kinetics, when accounting for the impact of the reverse rate. These results imply that Arrhenius analysis can access information of one rate-limiting transition state, e.g., of the Volmer or Tafel steps. However, these conditions are not broadly applicable. This has been understood early by Conway¹⁶⁻¹⁹, who introduced bias- and temperature-dependent charge transfer coefficients to expand the traditional Butler-Volmer equation into a more general free energy framework to capture entropic effects.

Acknowledging the great utility of the general free energy framework⁴³, in the text we refer to a “Butler-Volmer” regime if the bias-dependent kinetics show an approximately constant $\log A^\eta$. We chose this distinction not to diminish the great utility of the more generalized free energy framework, but to highlight that a large part of electrocatalyst literature typically (over)analyzes linear Tafel slopes at a constant temperature, i.e., assuming $\alpha_S = 0$. Similarly, it is broadly believed that $\alpha_H(\eta)$ (and $\alpha_S(\eta)$) should be bias-independent and that bias-dependent $\alpha_H(\eta)$ and $\alpha_S(\eta)$ are likely a reflection of non-kinetic mass transport effects^{44,45}. By restricting the analysis to linear Tafel fragments at a constant temperature, the whole breath of

inner-sphere kinetics is forced into an outer-sphere framework. Previously, it has been shown that bias-dependent coverage effects for the OER⁴⁶ can lead to non-linear Tafel slopes. Recently, we unveiled an important entropic and enthalpic impact of electric fields across the double layer and have obtained bias-dependent charge transfer coefficients that are a reflection of different kinetic regimes with bias²⁰⁻²³. Here, we show that the traditional Butler-Volmer equation and related Tafel analysis at a single temperature cannot be used to capture the complex multi-step ORR sequence, which is strongly impacted by a bias-dependent coverage, and electric field effects and potentially surface restructuring. In any case, it is not limited by one rate-limiting step across potentials. In fact, temperature-dependent ORR rates were previously studied with RDEs at low current densities on Pt¹⁴ and Ru/C-Nafion¹⁵ in concentrated phosphoric acid (25 - 250°C) and in 0.5M H₂SO₄ (25 – 75°C), respectively. Both ORR studies hypothesized that the rates are accelerated by a potential-dependent activation entropy, while others explored bias-dependent coverage effects^{10,11}. However, neither these nor our own study²⁰ resolved a turning point in the kinetics and the nature of the bias-dependent charge transfer coefficients.

36. Santos, E., Aradi, B., Van Der Heide, T. & Schmickler, W. Free energy curves for the Volmer reaction obtained from molecular dynamics simulation based on quantum chemistry. *Journal of Electroanalytical Chemistry* **954**, 118044 (2024).

37. Briega-Martos, V., Guzman-Soriano, R., Jiang, J. & Yang, Y. The (mis)uses of Tafel slope. *Nat Catal* **8**, 863–866 (2025).

38. Erdey-Grúz, T. & Volmer, M. Zur Theorie der Wasserstoff-Überspannung. *Zeitschrift für Physikalische Chemie* **150A**, 203–213 (1930).

39. Curtiss, L. A. *et al.* Temperature Dependence of the Heterogeneous Ferrous-Ferric Electron Transfer Reaction Rate: Comparison of Experiment and Theory. *J. Electrochem. Soc.* **138**, 2032 (1991).

40. Durst, J., Simon, C., Hasché, F. & Gasteiger, H. A. Hydrogen Oxidation and Evolution Reaction Kinetics on Carbon Supported Pt, Ir, Rh, and Pd Electrocatalysts in Acidic Media. *J. Electrochem. Soc.* **162**, F190 (2014).

41. Stühmeier, B. M., Pietsch, M. R., Schwämmlein, J. N. & Gasteiger, H. A. Pressure and Temperature Dependence of the Hydrogen Oxidation and Evolution Reaction Kinetics on Pt Electrocatalysts via PEMFC-based Hydrogen-Pump Measurements. *J. Electrochem. Soc.* **168**, 064516 (2021).

42. Sheng, W., Gasteiger, H. A. & Shao-Horn, Y. Hydrogen Oxidation and Evolution Reaction Kinetics on Platinum: Acid vs Alkaline Electrolytes. *J. Electrochem. Soc.* **157**, B1529 (2010).

43. Dickinson, E. J. F. & Wain, A. J. The Butler-Volmer equation in electrochemical theory: Origins, value, and practical application. *Journal of Electroanalytical Chemistry* **872**, 114145 (2020).

5. The authors claim that their results indicate that the reactant pressure electrochemically polarizes the transition state of a rate-limiting step in a manner analogous to applied overpotential. This is confusing as stated. I suppose the authors may be describing a pressure dependent effect on the structure of the electrochemical double layer which changes effective polarization, but it's quite speculative and unclear, especially at this point in the paper and probably deserves revisions as also described above in the discussion section

Answer: We thank the reviewer for spotting this. We have removed this sentence, at this point in the manuscript to avoid confusion.

The short summary graph in the intro on **Page 2** now reads:

We observe that the overpotential-dependent Arrhenius pre-exponential factor and apparent activation energy cascade through a series of rate-limiting steps and, potentially, transition states with varying degrees of rate control and which are closely linked to and limited by pseudo-capacitive processes at the solid-water interface.

6. The section titled Discussion is just a discussion of the pressured effects primarily. There's discussion and speculation mixed in across the entire paper, so distinctly labeling this discussion is not appropriate. Please provide a more descriptively accurate section title.

Answer: We thank the reviewer for this good suggestion. We now call this section

Considerations on a pressure dependent rate constant.

Furthermore, we went over the whole text again to be even more explicit about discussion of experimental findings and our hypothesis based on previous studies on this complex matter. Additionally, we cut down the last section on the pressure dependence further and highlight that higher pressure will be needed in the future to test our hypothesis rigorously. Please see the highlighted manuscript.

7. The authors write that the lack of substantial nonlinear Arrhenius behavior precludes the impact of temperature-dependent intermediate coverages or structural changes. I would be less definitive with the language in this sentence. It seems very difficult to rule out all those things.

Answer: We thank the reviewer for this important point. We have reworded this statement in the main and supplementary information.

Page 3

The lack of substantial non-linear Arrhenius behavior **reduces the likelihood** - within the studied temperature range – that significant temperature-dependent intermediate coverages, temperature-induced structural changes or a deformed transition state²⁶ **lead to pronounced temperature-dependent activation energies.** **Note, this does not preclude the existence of temperature-dependent structural or chemical changes that are relevant on much faster time scales at the highly dynamic active sites or other effects that could become relevant in a larger temperature range.**

Supplementary Note 1

Mass transport of membrane electrode assembly. The mass transport provided by gas diffusion electrodes (GDEs) in membrane electrode assemblies (MEA) **reduces the possibility** of pronounced temperature dependent mass transport limitations that could superimpose themselves onto the kinetic data in a manner to exactly produce Arrhenius linear fits with very high R² values (reaching 0.999 for Rh).

8. The authors invoke the language of a frustrated phase transition, which has a very precise meaning in physics community. It's not clear the exact meaning they have in this study. I'm not sure it's appropriate. This should be thoughtfully evaluated to use precise language and definitions.

Answer: We agree that the term is not particularly common in ORR research. The analogy to the topic in the physics community has been discussed previously by Robert Schlögl when discussing metastable, active OER sites that often emerge in the transition between two equilibrium/relaxed crystal structures. We have added additional explanation to clarify this.

Page 7-8

The turning potential between the compensation and “Butler-Volmer” region in Fig. 1d is important for the catalyst activity. For the OER on Ni(OH)₂²¹, we observed that the turning point **occurs right at a kinetically-driven phase transition between two crystal structures^{46,62,63}**, **which is closely related to a metastable active phase (frustrated phase generated during**

operation). This takes place in parallel, and might be linked, to the ordering in the interfacial hydrogen bond network^{49,50}. Across OER catalysts, the maximum E_A and $\log A$ were essentially independent of the loading and, instead, the activity was reflected in the overpotential where the maximum E_A and $\log A$ were reached²¹.

Page 11

For the OER, we were able to link the activation parameters for Ni(OH)₂ to structural and oxidation state phase transitions and interfacial water ordering observed with laser spectroscopy by others^{48,49}.

9. The authors include a highly critical statement regarding the use of DFT to calculate intermediate binding energies on metallic surfaces and predict rate determining steps. While this criticism is warranted due to the simplicity of those models, I would encourage the authors to be more matter-of-fact about their statements. For example, they state that one of the calculations was the most popular DFT studies ever performed. More precisely, it is one of the highest cited papers in the field of computational electrocatalysis. They also write that the authors proclaimed loudly that they have found the origin of the overpotential. However, that's also not a factual statement that belongs in a scientific paper. One can simply state that the authors and many others conclude that binding energies are primarily responsible for rate control through this analysis, but that that is inconsistent with the study here and indeed much of detailed experimental electrochemistry.

Answer: We sincerely apologize for this opinionated sentence. We have now edited the text to stay more factual.

Page 12

The ORR activity trend (high-to-low) of Pt/C, Ir/C, Ru/C and Rh/C is shown in **Figure 3d** and is neither fully dominated by $\log A^\eta$ nor E_A^η . Such behavior between catalysts was observed by others and us for the hydrogen evolution reaction in acid and in base^{23,67-69}. Whereas we only extract the apparent activation parameters, such behavior is difficult to reconcile with the BEP relationships that assume a constant activation entropy. In fact, our results clearly contrast findings reported in pioneering DFT electrocatalysis studies on the ORR⁷⁰. In the past, theoretical computational studies have often assigned the origin of the overpotential and rate control to a rate-limiting intermediate binding energy on a static metallic surface and predicted volcano activity plots. Further, from the data in Fig. 3d it is currently difficult to reconcile how the elegant scaling relationships⁷¹ between chemically-similar intermediates alone could be applied to explain the catalyst activity. Even neglecting the hypothesized solvation pre-step, the bias-dependent catalyst surfaces undergo substantial chemical and structural changes, some of which leave behind clear fingerprints in the apparent activation parameters, such as the maximum activation energy at low bias. In **Figure 3e** we plot the potential E_{RHE} where the maximum E_A^{RHE} are reached (Supplementary Figure 27) and observe that the potential barely changes with pressure, but appears pinned by pseudo-capacitive oxidation state changes at the water-solid interface (Figure 3c). Thus, we conclude that in all generality, for multi-step reactions, the rate limiting steps and transition states can change their degrees of rate control^{32,33} with bias. This needs to be accounted for in new theoretical work.

70. Nørskov, J. K. *et al.* Origin of the Overpotential for Oxygen Reduction at a Fuel-Cell Cathode. *J. Phys. Chem. B* **108**, 17886–17892 (2004).

71. Koper, M. T. M. Thermodynamic theory of multi-electron transfer reactions: Implications for electrocatalysis. *Journal of Electroanalytical Chemistry* **660**, 254–260 (2011).

10. Overall in the paper there are many typos and grammatical errors. Please carefully revise for the final submitted version.

We went over the whole text again to spot remaining typos and grammatical errors.